# Ecdysteroid Content and Therapeutic Activity in Elicited Spinach Accessions

**DOI:** 10.3390/plants9060727

**Published:** 2020-06-09

**Authors:** Jonathan Gorelick, Rona Hacohen Iraqi, Nirit Bernstein

**Affiliations:** 1Eastern R&D Center, Kiryat Arba 90100, Israel; 2Institute of Soil, Water, and Environmental Sciences, Volcani Center, Rishon LeZion 7505101, Israel; ronae3@gmail.com; 3The Robert H. Smith Faculty of Agriculture, Food and Environment, Hebrew University of Jerusalem, Rehovot 7612001, Israel

**Keywords:** spinach, ecdysteroid, elicitation, anabolic activity, protein synthesis

## Abstract

While spinach is an established nutritionally important crop, its medicinal value is not as well known. Spinach is rich in ecdysteroids, insect hormone analogs with a number of medicinal properties including anti-oxidative, anti-inflammatory and even anabolic activity. However, the potential of spinach as a medicinal plant has not yet been developed. In this study, the ecdysteroid content of spinach was optimized to increase its therapeutic value. Spinach seeds from various sources were grown under controlled hydroponic conditions and analyzed for ecdysteroid content and related anabolic activity. Variations in ecdysteroid content and the related anabolic activity were observed among spinach accessions. A selected variety, Spinacia oleracea cv. Turkey, was exposed to various physical and chemical elicitors to increase and stabilize ecdysteroid content. A number of elicitors, including methyl salicylate and mechanical damage, significantly increased ecdysteroid content and anabolic activity 24 h after exposure. The effect was transient and disappeared 48 h thereafter. Further work is needed to identify the most suitable germplasm and elicitation conditions for optimal ecdysteroid content.

## 1. Introduction

### 1.1. Spinach

The recognized nutritional benefits of spinach are well known, rich in many vitamins and minerals [1]. While spinach is an established nutritionally important crop, the lesser known medicinal value hidden in spinach makes it a crop with significant market growth potential. Spinach was shown to possess anti-oxidative, anti-proliferative and anti-inflammatory properties [2]. A spinach enriched diet conferred a neuroprotective effect on aging rats, improving both their learning capacity and motor skills [3]. In addition to these medicinal properties which have largely been attributed to flavonoids, spinach contains another type of biologically active compound, ecdysteroids [4].

### 1.2. Ecdysteroids

Ecdysteroids, polyhydroxylated ketosteroids with long carbon side chains, are produced primarily in arthropods and plants, but are also present in fungi and even in marine sponges [5]. While 20-hydroxyecdysone (20HE) is the primary ecdysteroid found in arthropods, over 200 different ecdysteroids have been discovered in plants [6]. Although they are found throughout the living world, their discovery was only made within the last 60 years, and at first, their significance was not appreciated. In insects, ecdysteroids are involved in the molting process (ecdysis), while in plants there is evidence that they are involved in the plant defense response to insect herbivory [7]. The effects of ecdysteroids on mammalian cells were first documented in 1961 [8], and over time, the accumulation of data strongly supports their therapeutic activity [9]. Phytoecdysteroids are acknowledged today to possess a wide range of potentially therapeutic properties for humans, including anabolic [10], adaptogenic [11], hepatoprotective [12], spasmolytic [13] and anti-inflammatory activity [14].

Over 100 different preparations containing ecdysteroids can be found on the market containing either crude plant extracts or purified extracts with defined ecdysteroid content [15]. Most, which are sold over the Internet, are marketed to athletes, specifically body builders. Although mainly suggested to increase muscle, ecdysteroids are also sold to treat diabetes, to increase energy or as aphrodisiacs [16]. Some are even advertised for animals, including horses and dogs. In addition. ecdysteroids are also present in some cosmetics designed to improve moisture in the skin (Phenomen and Hydrastar from C. Dior).

### 1.3. Ecdysteroids in Spinach

Well known foods may contain lesser known therapeutic activities. Spinach (*Spinacia oleracea*) has been known to contain ecdysteroids, primarily 20HE, for some time [17], but is not widely used therapeutically as a source of ecdysteroids. The amounts present are considerably lower than in many of the other known ecdysteroid containing plants. However, of the major food crops, spinach contains the highest amounts of ecdysteroids, about 0.01% fresh weight [4]. Other related plants, including quinoa and asparagus, also contain significant levels as measured using High Pressure Liquid Chromatography (HPLC) and radioimmunoassay (RIA) [18]. Since spinach has a long history of use as a food crop and is generally recognized as safe, it is an ideal species to study the therapeutic activity of ecdysteroid enriched foods.

However, the ecdysteroid content found in spinach is quite variable [19]. This variation can be attributed to two main factors: genetic and environmental variability. The potential of genotype to affect ecdysteroid content was documented. Fifteen accessions of spinach grown under controlled conditions displayed significant differences in ecdysteroid content [20]. However, no wide-scale selection was undertaken for higher ecdysteroid producing lines. The other major factors involved in ecdysteroid content are the environmental conditions. Although many different variables may affect ecdysteroid content, its role in defense has focused research on defense related parameters primarily against insect pests [21]. Mechanical damage to the roots or leaves elicited increased ecdysteroid content [22]. Both exogenous 20HE and plant produced ecdysteroids produced abnormal molting, immobility, reduced invasion, impaired development and death in insects [21]. Production of 20HE in spinach is elicited by both mechanical wounding and insect feeding [21]. Phytoecdysteroids were found to protect spinach from plant-parasitic nematodes and may confer a mechanism for nematode resistance [7]. However, the response of spinach to different biotic or abiotic elicitors with regards to its therapeutic activity has not been characterized. Therefore, the goal of the present study was to characterize the ecdysteroid content and activity of a number of different spinach accessions grown under controlled conditions in order to evaluate the natural chemical variation present in varying spinach genotypes and their associated anabolic activity. Subsequently, the environmental role in chemical variation was evaluated by determining the effects of elicitation on ecdysteroid production. Elicitors induce physiological responses in the plant cells, which in turn often induce alterations to secondary metabolism. The effects of exposure of the plants to the elicitors on physiological parameters of the plant tissue was therefore tested to evaluate potential stress responses. The elicitors selected are known to stimulate the production of secondary metabolites in plants. Methyl salicylate (MS) and chitosan have been shown to be very effective at inducing the plant defense response [23,24]. MS is a natural plant derivative of salicylic acid implicated as an airborne signal involved with systemic acquired resistance [25]. Chitosan is a polysaccharide produced from the breakdown of the fungal cell wall component chitin. Treatment with chitosan mimics an attack from a fungal pathogen which activates the plant defense response [26]. Jasmonic acid and its related methyl ester, methyl jasmonate (MJ), play a key role in the signal transduction pathways which regulate the plant defense response and are involved in the production of various bioactive secondary metabolites [27]. Mechanical damage to either aerial or root tissue is known to activate the plant defense response and stimulate the production of secondary metabolites [28]. Membrane leakage and osmotic potential were measured as indicators of membrane damage and water relation status, respectively and concentration of photosynthetic pigments as the potential for carbon fixation.

## 2. Results

### 2.1. Ecdysteroid Content

The content of 20-HE was analyzed in 7 selected spinach types grown under controlled conditions (Figure 1a). Interestingly, there was a range of concentrations found in the leaf material from the various spinach sources. The accessions with the highest concentrations were PI604787 and P175312, containing an average of 17.3-µg/g 20HE and 16.8-µg/g 20HE, respectively. These accessions also displayed a very large range of concentrations among the different plants from 10.1–25.8-µg/g 20HE and 8.2–27.0-µg/g 20HE, respectively. Spinach leaves from the Turkey variety contained less 20HE (12.3 µg/g) although this decrease was not significant. The lowest concentrations of 20HE were found in PI169673 and NLS6085, containing 10.27-µg/g 20HE and 9.3-µg/g 20HE, respectively. Interestingly, the range of concentrations of 20HE found in each spinach type also varied. The spinach types with higher 20HE content tended to have greater variation between plants while the types with lower concentrations had less variation in 20HE content.

### 2.2. Anabolic Activity

The anabolic activity associated with ecdysteroids was analyzed using a cellular model of protein synthesis in skeletal muscle cells. All spinach accessions tested significantly stimulated leucine incorporation compared to untreated cells (4662 ± 304 Decays per minute (DPM)). The different spinach types produced varying increases in protein synthesis at least partially correlating with 20HE concentration (Figure 1B). PI175312, with a 20HE content of 16.8 µg/g, produced the greatest increase in protein synthesis, while NLS6085, with a 20HE content of 9.3 µg/g stimulated the smallest increase. Interestingly, the effects on protein synthesis could only partially be correlated to ecdysteroid content. The highest increase in protein synthesis was produced by the Turkey variety, even though its 20HE content was lower than PI604787, which contained the highest content of 20HE. While Genesis 8413 and PI370602 contained similar 20HE content (11.9–11.5 µg/g) PI370602 increased protein synthesis significantly more than Genesis 8413. PI6973, which contained slightly less 20HE (10.3 µg/g) generated a slightly less increase, although the difference was not statistically significant. Surprisingly, PI604787, which contained comparable 20HE with PI75312, stimulated a significantly smaller increase in protein synthesis.

### 2.3. Elicitation on Ecdysteroid Content

After verifying the ecdysteroid content and the associated biologic activity, elicitation was utilized to increase and stabilize the ecdysteroid content. Because of its potent activity on protein synthesis—as well as the fact that it is a modern developed variety, possessing many agriculturally desirable attributes in addition to high ecdysteroid content—the Turkey variety was selected for use in elicitation experiments.

All elicitors studied produced a change in 20HE content (Figure 2A). The greatest increase in 20HE content was observed in plants treated with MS or root or leaf-wounding. Twenty-four hours after initiation of the MS treatments, a ~200% increase in 20HE from 8.3 to 24.1–24.7 µg/g was observed. MJ treatment increased 20HE content by ~100% reaching 17.2 µg/g. The observed increase was transient. After 72 h, not only did the previously observed increase in 20HE content dissipate, but it even dropped to about half of the content found in the control untreated plants. Surprisingly, chitosan treatment did not significantly increase 20HE content.

### 2.4. Elicitation and Anabolic Activity

In order to correlate the changes in ecdysteroid content observed in elicited spinach with the associated therapeutic activity, effects on protein synthesis in skeletal muscle were studied. Skeletal muscle cells (L6) were treated with extracts from leaves of spinach plants treated with various elicitors and the incorporation of tritiated leucine into protein was quantified (Figure 2B).

Supporting our previous findings, leaves from untreated spinach leaves significantly increased protein synthesis in skeletal muscle cells compared with cells which received the vehicle (6485 ± 476.6 DPM). Furthermore, 24 h after elicitation, plants stimulated a significantly larger increase in protein synthesis than that produced by untreated plants. The largest increase in protein synthesis was produced by plants treated with wounding of leaf or roots as well as MS. A smaller yet significant increase in protein synthesis was observed in MJ treated plants compared with non-elicited plants. This increase dissipated after 72 h, reverting to activity similar to the protein synthesis observed in non-elicited plants. In the case of root-wounding, after 72 h the stimulated protein synthesis was actually lower, although not statistically significant, than in non-elicited plants. These changes in anabolic activity, both the increase after 24 h and the subsequent decrease after 72 h, parallel the previously observed increases in 20HE content. The increase in protein synthesis from chitosan treated plants was similar to non-elicited plants.

### 2.5. Physiological Response to Elicitation

Evaluating changes in stress-related physiological parameters was aimed at identifying plant characteristics that correlate with stimulation of increased bioactivity. The elicitation treatments did not affect membrane leakage from the leaf tissue (Figure 3A), nor the osmotic potential of the tissue sap (Figure 3B). While many changes in secondary metabolite production are result of the physiological stress which may be induced in the plant by the elicitors, this does seem to be the case in the present study. The lack of change in membrane leakage or osmotic potential suggest that the elicitation treatments which increased ecdysteroid content after 24 h did not produce significant physiological stress in the plants.

Almost no changes in pigmentation were observed following elicitation treatments (Figure 4). Only chitosan treatment significantly reduced both chlorophyll a and b while increasing carotenoid levels. This is surprising as most studies using chitosan report an increase in chlorophyll content [29,30]. While other elicitors like salicylic acid have been observed to lower chlorophyll content [31], the observed chlorophyll lowering effects of chitosan have never been reported. Surprisingly, while salicylic acid treatment has been shown to increase carotenoid content in plants [32], the effects of chitosan on carotenoid levels have not been documented.

## 3. Discussion

A range of 20HE concentrations was observed in the various spinach material with sources with higher 20HE content possessing greater interplant variation. This increased chemical variability among different plant populations have been previous documented for a number of species [33,34,35]. These findings suggest that the level of variation in ecdysteroid content within a single population may vary depending on the germplasm. In addition. different levels of genetic heterogeneity may exist among the different accessions studied.

The anabolic activity associated with ecdysteroids was confirmed with the different spinach types producing varying increases in protein synthesis. Interestingly, the anabolic activity did not completely correlate with ecdysteroid content. It is quite plausible that other non-ecdysteroid compounds are present in spinach with anabolic activity [2]. In addition. there may also be compounds which inhibit anabolic activity. Further work is needed to better characterize the various chemical factors influencing spinach’s anabolic activity.

Overall, we have confirmed that the 20HE content in spinach varies greatly among different germplasm. Of course, a larger survey is needed to adequately evaluate the 20HE variation among spinach worldwide. The ability of elicitation to modulate 20HE content was also observed, supporting previous findings [22]. While the ability to significantly improve the ecdysteroid content in spinach has been supported, there is still much work ahead. The development of a spinach based ecdysteroid-containing nutraceutical demands additional biotechnological research. Optimization of ecdysteroid content must be monitored using sophisticated chemical analysis as well as cell based biologic assays in order to assure the therapeutic activity.

## 4. Materials and Methods

### 4.1. Plant Material

Spinach seeds, Spinacia oleracea cv. Turkey and cv. Sabanach (Genesis 8413) were obtained from Hishtil (Nehalim, Israel) and Genesis Seeds, Ltd. (Rehovot, Israel) accordingly. Spinach seeds of NSL 6085 (U.S.), PI 169673 (Turkey), PI 175312 (India), PI 370602 (Macedonia), PI604787 (Afghanistan) were obtained from the U.S. Department of Agriculture’s Agricultural Resource Service, Germplasm Resources Information Network (USDA-ARS GRIN). Accessions were selected from the USDA seed bank based on reported insect resistance with the likelihood that their ecdysteroid content may be naturally higher [36].

### 4.2. Growing Conditions

All plants were germinated from seed at a commercial nursery (Hishtil, Nehalim, Israel) and after 24 days, transferred to aerated one-quarter strength modified Hoagland hydroponic solution [37], 5 plants each into 14 L containers in a climate controlled greenhouse maintained at 21–30 °C at the Volcani Institute (Rishon Le-Ziyon, Israel) [38]. Iron (Fe-EDTA) was added to the growing media to a final concentration of 50-µM and sodium (as NaCl) to a final concentration of 1 mM. The pH of the solution was maintained at 5.7 through daily adjustment with KOH. The hydroponic solution was replaced weekly.

### 4.3. Elicitation

Elicitors including methyl salicylate (MS), methyl jasmonate (MJ) and chitosan—as well as leaf- or root-wounding—were utilized to stimulate the plant stress response to increase the 20HE content and anabolic activity of spinach based on [39] with minor modifications.

Spinach seedlings grown under the previously described hydroponic conditions were divided using a random block design into five replicated hydroponic containers of five plants each per treatment (25 plants per elicitation treatment). Selected elicitor concentrations were determined based on preliminary studies (results not shown). The three chemical elicitors (390 µg/L MS, 390 µg/L MJ, 150 mg/L chitosan) were added to the hydroponic nutrient media. For plants sampled after 72 h, the 24 h treatment with the chemical elicitors was followed by replacement with the original hydroponic growing solution for an additional 48 h. Physical elicitation was produced using scissors to generate slight cuts in the upper leaves for leaf-wounding. Root-wounding was produced by gentle pruning of the root tips. At the conclusion of the 24 or 72 h periods, plants were harvested, and samples analyzed for physiological response, chemical composition and bioactivity. The physiological response of elicited plants was compared to control plants by measuring the membrane leakage and osmotic potential of leaf sap [40].

### 4.4. Membrane Leakage

Measurement of membrane leakage—an indicator of stress-related injury [41]—were studied as previously described with minor changes [42]. After leaf segments were shaken in 30 mL of double-distilled water for 24 h, electrical conductivity (EC) was measured using a conductivity meter (CyberScan CON 1500; Eutech Instruments, Ayer Rajan Crescent, Singapore). Leaf pieces were subsequently autoclaved and shaken for 1 h. Conductivity measurements were repeated and the ratio between the first and the second conductivity measurements were used to calculate membrane leakage, presented as percentages. Five replicated leaves were averaged to generate the final results.

### 4.5. Osmotic Potential

For osmotic potential measurements, leaf tissue from the center of the youngest mature leaf on the plant was sampled—and following washing with deionized water—was blotted dry and frozen in liquid nitrogen in 1.5 mL micro-tubes and stored at 5 °C for further analyses. The frozen tissue was crushed with a glass rod, the bottom of the tubes pin-pricked, set inside another 1.5 mL tube and centrifuged in a refrigerated centrifuge (Sigma Laboratory Centrifuges, Osterode am Harz, Germany) at 5 °C at 10,000 rpm for 4 min. The fluid collected in the lower micro-tube (100 L) was used for measurement of osmotic potential using a cryoscopic micro-osmometer (Osmette; Precision Systems, Natick, MA, USA), measuring the sap freezing point. Results are presented in mOsm * kg H_2_O^−1^. Five replicates were analyzed from each treatment.

### 4.6. Chlorophyll and Carotenoids Content

The youngest mature leaf from each plant was separated and rapidly washed in distilled water. A tissue segment of 20 × 20 mm removed from halfway along the length of the leaf was used for analyzing chlorophyll a and b and carotenoid content. Discs of 6 mm diameter, cut from leaf sections avoiding the mid-rib, were placed in 80% (*v/v*) ethanol and heated for 30 min at 92 °C. After boiling, the soluble extract was collected in 2 mL micro tubes. Extraction was repeated with the remaining tissue at room temperature for 15 min and the combined extracts was vortexed. Absorbance was measured at 663, 646 and 470 nm using a Genesys 10 UV Scanning spectrophotometer (Thermo Scientific, Madison, WI, USA). Chlorophyll a and b and carotenoid content was calculated based on [43].

### 4.7. Ecdysteroid Extraction and Quantification

The youngest fully mature leaf on the plants was sampled for analysis of ecdysteroid content and immediately frozen and stored at −80 °C. In experiments were the concentration was also analyzed 48 h after the removal of the elicitor, i.e., 72 h after the initiation of the elicitation treatment, the same leaf was sampled for the analyses. The collected samples were ground and extracted in methanol for 24 h. The supernatant was filtered and evaporated to dryness under reduced pressure. Samples were resuspended in 70% methanol in water and then partitioned against hexane. The aqueous fraction was evaporated, resuspended in water, and partitioned against butanol. The butanol fraction was evaporated, resuspended in methanol, filtered (0.45 µm), and analyzed using HPLC (Waters LC Module 1 with 996 PDA detector, Milford, MA, USA). Samples (20 µL) were injected onto a ODS column (Phenomenex Luna (4.6 mm • 25 cm, 5 µm), Torrance, CA, USA) using an isocratic elution with a mobile phased of water/methanol (40:60) and a flow of 1.0 mL/min. Quantification was performed using standard curves generated with >93% purified 20-hydroxyecdysone (1–1000 ng) (Sigma-Aldrich, St. Louis, MO, USA).

### 4.8. Protein Synthesis Activity

The effects of partially purified spinach extracts on total protein synthesis in skeletal myotubes were studied based on [10]. Briefly, L6 cells were washed with serum-free Dulbecco’s modified Eagle’s medium (DMEM) and treated with purified spinach extracts, insulin like growth factor 1 (IGF−1) or vehicle, 0.1% ethanol, four wells per treatment. Samples were added to serum-free medium containing 5-µCi/mL [3H] Leucine. Cells were incubated for 4 h before protein measurement. Following treatment, cells were washed with cold phosphate-buffered saline (PBS), followed by the addition of 5% trichloroacetic acid (TCA) to precipitate protein. After 30 min at 4 °C, the TCA was removed, and the precipitate was dissolved in 0.5-M NaOH (500 µL). The dissolved precipitate (400 µL) was added to scintillation vials with 5 mL of scintillation fluid (Ready Safe, Beckman Coulter, Fullerton, CA, USA). Decays per minute (DPM) were measured in a liquid scintillation counter (LS 6500, Beckman Coulter, Fullerton, CA, USA). Total protein was quantified using the bicinchoninic acid (BCA) method following the manufacturer’s instructions (Pierce, Rockford, IL, USA). Data were expressed as DPM per milligram of total protein. Each experiment was performed in triplicate. The results were expressed as mean ± SEM. Statistical significance was determined using one-way ANOVA and Tukey’s HSD for comparison of means.

## Figures and Tables

**Figure 1 plants-09-00727-f001:**
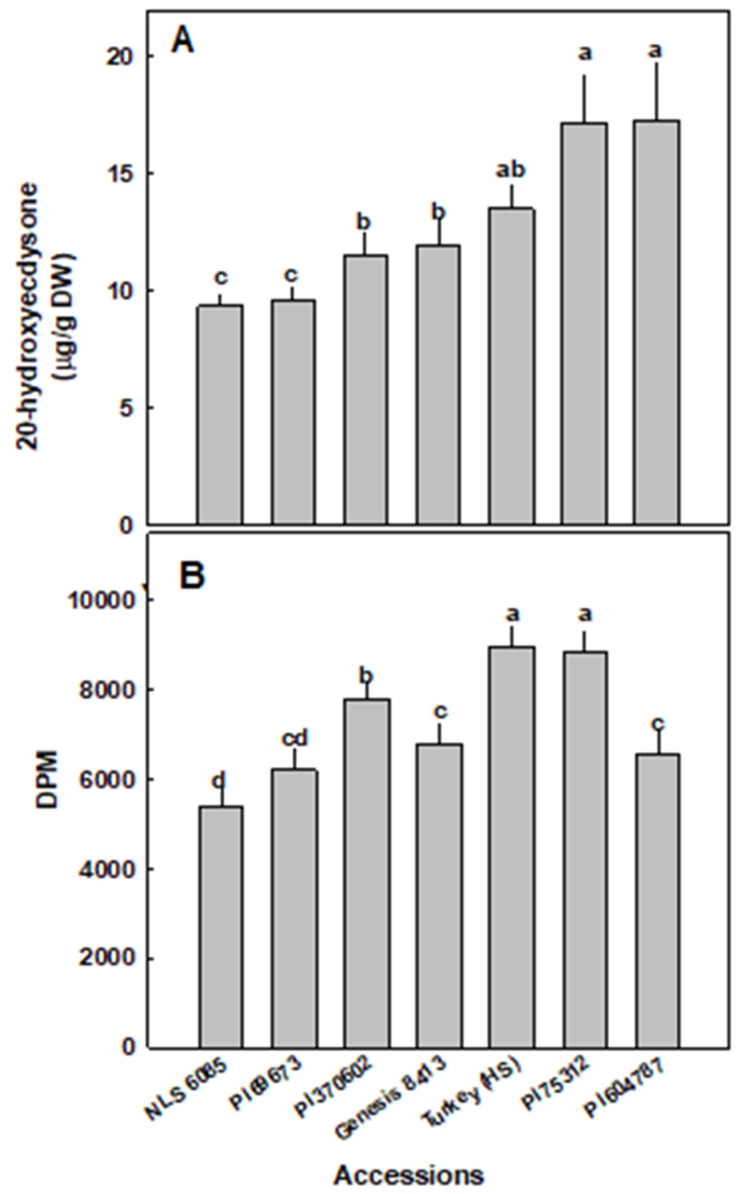
20-Hydroxyecdysone content and anabolic activity in the leaves of 7 selected spinach accessions. (**A**) 20HE content was quantified using partially purified extracts of leaves from each of the 7 spinach accession analyzed via HPLC; (**B**) [3H] Leucine incorporation in L6 myotubes treated with different ecdysteroid-containing spinach extracts for 4 h was measured. Decays per minute (DPM) was normalized by total protein. The data represent the mean values ± standard error of the mean (SEM) of four replicates. Different letters above the means represent significant difference based on Tukey’s honestly significant difference (HSD) test at α = 0.05.

**Figure 2 plants-09-00727-f002:**
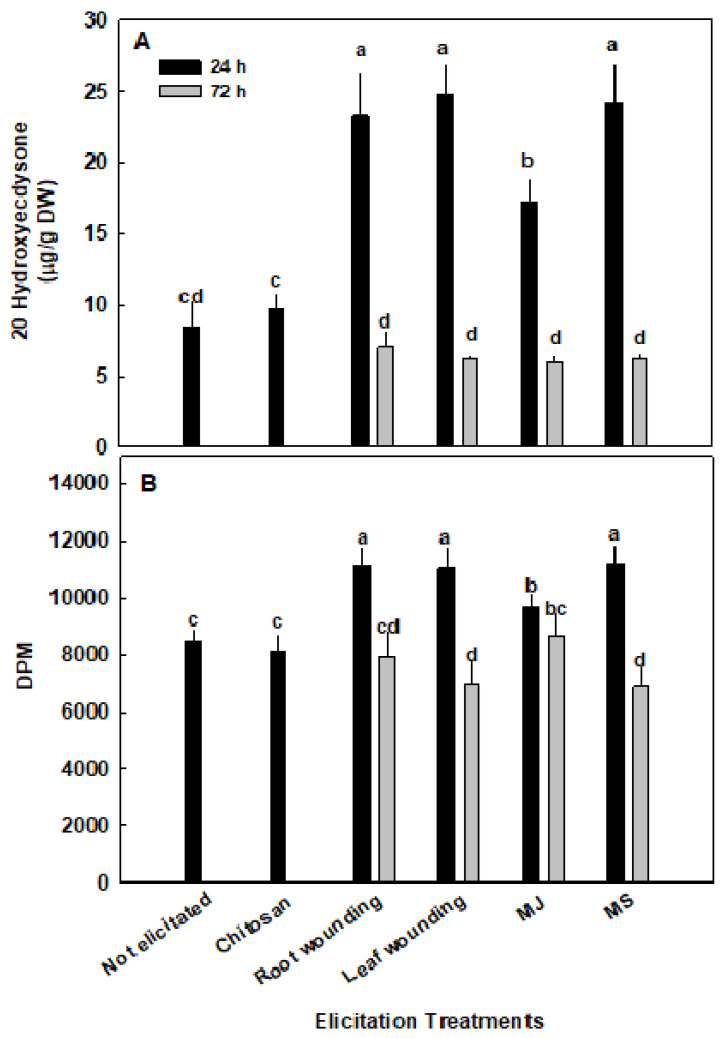
20HE content and anabolic activity in elicited Turkey variety spinach plant leaves. (**A**) 20HE content was quantified using partially purified extracts of leaves from elicited spinach analyzed via HPLC; (**B**) [3H] Leucine incorporation in L6 myotubes treated with elicited spinach extracts for 4 h was measured. Decays per minute (DPM) was normalized by total protein. The data represent the mean values ± standard error of the mean (SEM) of four replicates. Different letters above the means represent significant difference based on Tukey’s honestly significant difference (HSD) test at α = 0.05.

**Figure 3 plants-09-00727-f003:**
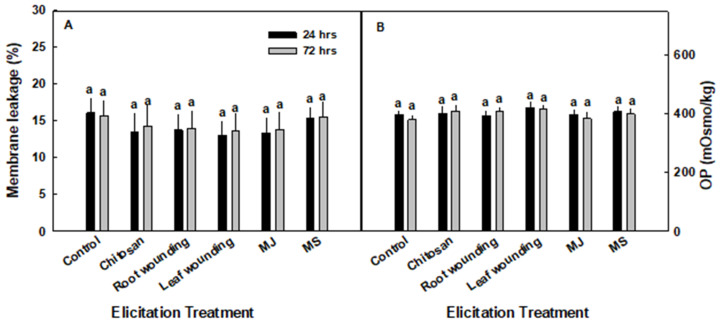
(**A**) Membrane leakage and (**B**) osmotic potential in elicited Turkey variety spinach plant leaves. The data represent means ± SE (*n* = 5). Different letters above the means represent significant difference based on Tukey’s HSD test at α = 0.05.

**Figure 4 plants-09-00727-f004:**
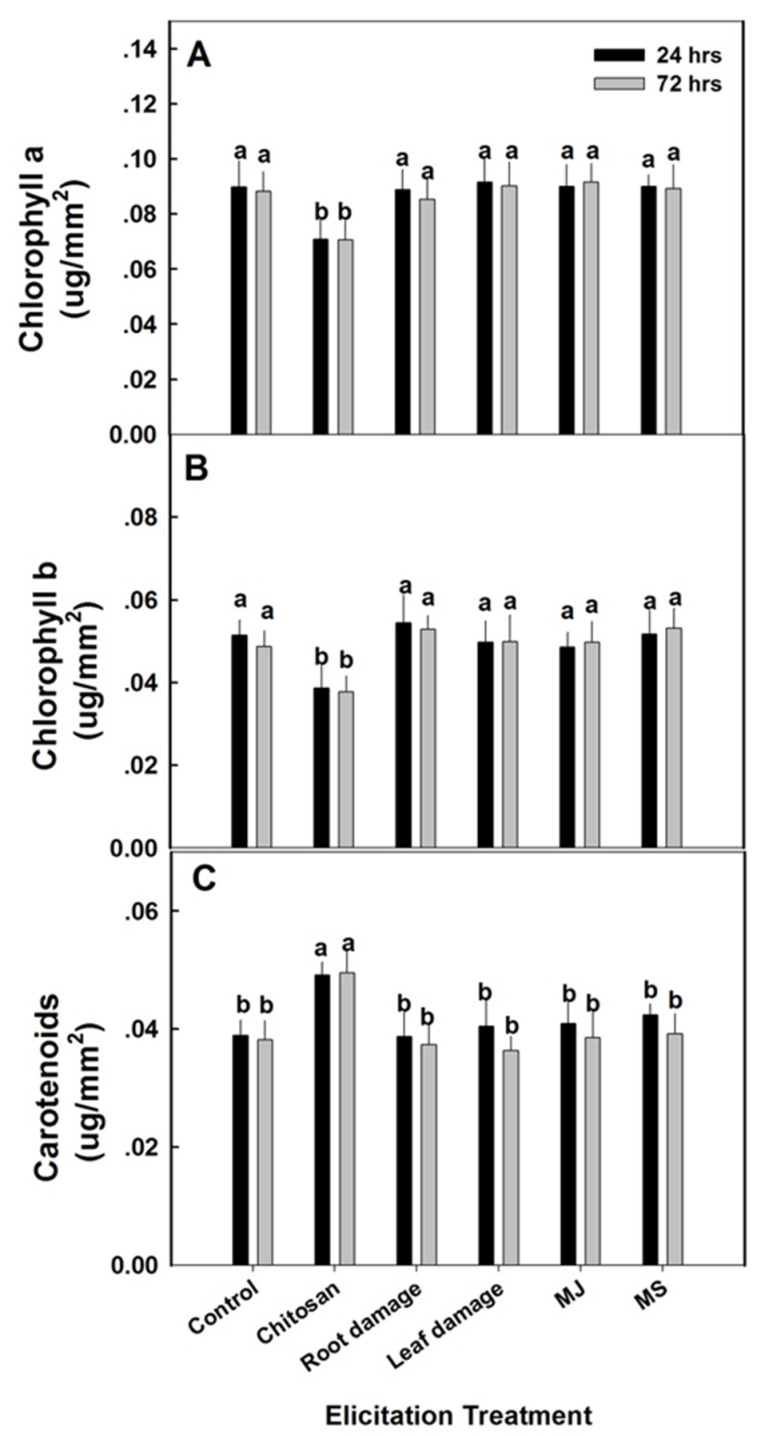
Photosynthetic pigments in elicited Turkey variety spinach leaves. Measurements of (**A**) chlorophyll a, (**B**) chlorophyll b, and (**C**) carotenoids were conducted 24 h after the initiation of the elicitation treatments (24 h) and 48 h thereafter (72 h). Methyl salicylate (MS), methyl jasmonate (MJ) and chitosan were removed after 24 h exposure. The data represent means ± SE (*n* = 5). Different letters above the means represent significant difference based on Tukey’s HSD test at α = 0.05.

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
