# Peer review of "Ecdysteroid Content and Therapeutic Activity in Elicited Spinach Accessions"

_plants, 2020, doi:10.3390/plants9060727_

Round 1
Reviewer 1 Report
In the introduction It seems inappropriate to add subtitle (1.1 1.2 1.3) however if this is permitted by “Plants” instructions it can be considered.
Some text in the introduction needs references please check and add a reference when needed.
The reference list should be updated with recent papers in the field.
In figure 1B in abscise axis and explain DPM (Decays per minute). The same in other figures please explain abbreviations to make the figure comprehensible from its legend: there is no need to search in the text.
In whole manuscript any abbreviation should be defined in first appearance in the text (ex. DPM).
In fig 2. A please correct the unit µg/gram to µg/g to make it uniform with previous text.
Please add a statistical method section; to explain the experiment design, replicates, the statistical software and methods for comparing varieties and experimental conditions.
The discussion does not include any reference!? Maybe it is the conclusion. The paper was not well discussed with previous literature, the conclusions were not clear: please add few sentences showing the main outcomes of this study and its potential application (compared to the literature).
Author Response
Response to Reviewer 1 Comments
Point 1: In the introduction It seems inappropriate to add subtitle (1.1 1.2 1.3) however if this is permitted by “Plants” instructions it can be considered.

Response 1: We have removed the subtitles from the introduction.
Point 2: Some text in the introduction needs references please check and add a reference when needed.
Response 2: We have added a number of references to the introduction
Point 3: The reference list should be updated with recent papers in the field.
Response 3: We have added citations from more recent papers, although the majority of work on this topic was not recently published.
Point 4: In figure 1B in abscise axis and explain DPM (Decays per minute). The same in other figures please explain abbreviations to make the figure comprehensible from its legend: there is no need to search in the text.
Response 4: An explanation for the acronym DPM was added were relevant.
Point 5: In whole manuscript any abbreviation should be defined in first appearance in the text (ex. DPM).
Response 5: Explanations for first appearances for all abbreviations were added.
Point 6: In fig 2. A please correct the unit µg/gram to µg/g to make it uniform with previous text.
Response 6: The units were corrected as requested.
Point 7: Please add a statistical method section; to explain the experiment design, replicates, the statistical software and methods for comparing varieties and experimental conditions.
Response 7: A statistical analysis section (4.9) was added to the Materials and Methods.
Point 8: The discussion does not include any reference!? Maybe it is the conclusion. The paper was not well discussed with previous literature, the conclusions were not clear: please add few sentences showing the main outcomes of this study and its potential application (compared to the literature).
Response 8: References were added to the discussion as a well as conclusions regarding the main outcomes and potential applications.
Reviewer 2 Report
Dear Authora,
please, find all my comments and suggestions within the .pdf document. In order to see the comments, please use Adobe pdf reader, or PDF XChange Viewer as it probably will not be visible in Edge.
My general comments are this:
- Try to better explain the goals of the research: you have not investigated just the content of ecdysteroids in dofferent accessions, but also different phisiological response like catabolic reaction etc. Try to be more precise about the explanation of the goals.
- Some centences should be reformulated (see the comments).
- There are sone strange signs within the manuscript; maybe it is just due to converting the paper into pdf.

Author Response
Response to Reviewer 2 Comments
Point 1: Try to better explain the goals of the research: you have not investigated just the content of ecdysteroids in dofferent accessions, but also different phisiological response like catabolic reaction etc. Try to be more precise about the explanation of the goals.
Response 1:. The goals of this work were elaborated.
Point 2: Some centences should be reformulated (see the comments).
Response 2: Selected sentences were edited.
Point 3: There are sone strange signs within the manuscript; maybe it is just due to converting the paper into pdf.
Response 3: We are not clear on the comment as we did not see any strange signs.
Reviewer 3 Report
The selected plant species and the topic is good but this manuscript is poorly written. Introduction should be more focused and detailed in order to understand the reader the significance of this topic. Clear focus is needed. I think authors should have been describe the selected spinach types and their origin because it can affect their secondary metabolite production. Why did authors choose the 24 h and 72 h for treatment? The treatments of elicitation in my opinion is not successful and not provide evidence of the importance of 20HE in spinach.
Author Response
Point 1: The selected plant species and the topic is good but this manuscript is poorly written. Introduction should be more focused and detailed in order to understand the reader the significance of this topic. Clear focus is needed. I think authors should have been describe the selected spinach types and their origin because it can affect their secondary metabolite production.

Response 1: We have edited the introduction in order to clearly explain the goals. The selected spinach type were partially described as possessing reported insect resistance with the hope that their ecdysteroid content may be naturally high.
Point 2: Why did authors choose the 24 h and 72 h for treatment?
Response 2: The selected time points were used based on preliminary studies of the the longevity of the observed elicitation.
Point 3: The treatments of elicitation in my opinion is not successful and not provide evidence of the importance of 20HE in spinach..
Response 3: While the successfulness of elicitation remains to be determined, the importance of 20HE in spinach is clear with a number of prior studies documenting the ecological and biochemical roles.
Reviewer 4 Report
Detailed comments:
L29: change mineral by minerals
L43-47: requires more references
L48: consider revision: inverse prepations & ecdysteroids ; add a reference
L51-52: there is no subject in the sentence
L51-54 add references such as: https://www.nature.com/articles/srep37322
L57: 20HE: add complete name of the molecule for the first time you cite it in abbreviation
L60-61: Add reference
L56-59: consider revision, contradictory information provided, not clear
L64-65: add reference
L69: replace is by are
L70-71 Explain better "defense related parameters", add references
L74 add reference
L86 Add Standar error to the mean values
Fig1 & 2 consider expliting it in 4 different ratios, as the explanations for the b parts are too far from the figure there is no easy to understand the relationship of the two graphics together.
Fig 1.b explain in the legend what is DPM
L110-111: there is no test of the correlation
L119-123: this justification should be located in the Material and Methods section
L124-133: This paragraph has to be placed in the introduction, no place for it in the results
L135: specify what is MS and later MJ the first time cited
L167-170: this justification should be located in the Material and Methods section
L197: consider revision of the sentence, not clear
L197-L212: Discussion requires litterature support in fact is more a conclusion than a discussion. Conclusion is missing.
L233: change format of the reference
L335 supress each
L239 change was by were
L240 supress "an" before additional
Around L274 there are several times a symbol that has replaced several simbols such as -80°C.
In each Machine used it's required to add the city and the country of origin
L284: at least for the standards it's necessary to add the company and the degree of quality of the standard
Author Response
Point 1: L29: change mineral by minerals
Response 1: Corrected.
Point 2: L43-47: requires more references
Response 2: We have added a number of references to the introduction
Point 3: L48: consider revision: inverse prepations & ecdysteroids ; add a reference.
Response 3: The sentence was revised and a reference added.
Point 4: L51-52: there is no subject in the sentence
Response 4: Sentence was removed.
Point 5: L51-54 add references such as: https://www.nature.com/articles/srep37322
Response 5: References were added.
Point 6: L57: 20HE: add complete name of the molecule for the first time you cite it in abbreviation
Response 6: The complete name is present the first time 20HE is mentioned.
Point 7: L60-61: Add reference
Response 7: Reference was added.
Point 8: L56-59: consider revision, contradictory information provided, not clear
Response 8: We have revised this section.
Point 9: L64-65: add reference
Response 9: Reference was added.
Point 10: L69: replace is by are
Response 10: The sentenced was corrected
Point 11: L70-71 Explain better "defense related parameters", add references
Response 11: “Denfence related parameters” was explained and a reference added.
Point 12: L74 add reference
Response 12: Reference was added.
Point 13: L86 Add Standar error to the mean values
Response 13: Standard Error of the Mean appears in Fig. 1.
Point 14: Fig1 & 2 consider expliting it in 4 different ratios, as the explanations for the b parts are too far from the figure there is no easy to understand the relationship of the two graphics together.
Response 14: We considered a number of potential ways to display the data, with each way having pros and cons.
Point 15: Fig 1.b explain in the legend what is DPM
Response 15: An explanation was added.
Point 16: L110-111: there is no test of the correlation
Response 16: The data were subjected to one-way analysis of variance (ANOVA) followed by Tukey’s HSD test.
Point 17: L119-123: this justification should be located in the Material and Methods section
Response 17: The justification was placed in the Results section in order to explain why the following results used this variety.
Point 18: L124-133: This paragraph has to be placed in the introduction, no place for it in the results
Response 18: The paragraph was moved to introduction
Point 19: L135: specify what is MS and later MJ the first time cited.
Response 19: Abbreviations were explained.
Point 20: L167-170: this justification should be located in the Material and Methods section
Response 20: Justification was moved to Materials and Methods.
Point 21: L197: consider revision of the sentence, not clear
Response 21: Sentence was revised.
Point 22: L197-L212: Discussion requires litterature support in fact is more a conclusion than a discussion. Conclusion is missing.
Response 22: The discussion was elaborated with references added.
Point 23: L233: change format of the reference
Response 23. We are not clear on which reference
Point 24: L335 supress each
Response 24: We do not understand this point.
Point 25: L239 change was by were
Response 25: The sentence states “…plants were harvested”
Point 26: L240 supress "an" before additional
Response 26: The sentence requires the indefinite article, an.
Point 27: Around L274 there are several times a symbol that has replaced several simbols such as -80°C.
Response 27: the micro symbol, m seems to become corrupted from the text.
Point 28: In each Machine used it's required to add the city and the country of origin
Response 28: City and country was added for each instrument mentioned.
Point 29: L284: at least for the standards it's necessary to add the company and the degree of quality of the standard
Response 29: Company and percent purity were added.
Round 2
Reviewer 1 Report
The current version of the paper is well revised by the authors according to previous comments. I have no further comment to the authors. The paper can be considered for publications in the current form.
Reviewer 4 Report
Corrections asked have been made and special attention has been made on the references and results presentation.
Much more coherence has been given to the document